

# A novel approach to immunoapheresis of C3a/C3 and proteomic identification of associates

Wolfgang Winnicki[1], Peter Pichler[1], Karl Mechtler[2], Richard Imre[2], Ines Steinmacher[2], Gürkan Sengölge[1], Daniela Knafl[1], Georg Beilhack[1] and Ludwig Wagner[1]

[1] Department of Medicine III, Division of Nephrology and Dialysis, Medical University of Vienna, Vienna, Austria
[2] ProtChem Facility, Research Institute of Molecular Pathology, Vienna, Austria

## ABSTRACT

**Background:** Complement factor C3 represents the central component of the complement cascade and its activation split product C3a plays an important role in inflammation and disease. Many human disorders are linked to dysregulation of the complement system and alteration in interaction molecules. Therefore, various therapeutic approaches to act on the complement system have been initiated.

**Methods and Results:** Aiming to develop a tool to eliminate C3a/C3 from the circulation, in a first step a high affine murine monoclonal antibody (mAb) (3F7E2-mAb) was generated against complement factor C3 and selected for binding to the C3a region to serve as immunoaffinity reagent. Functional testing of the 3F7E2-mAb revealed an inhibition of Zymosan-induced cleavage of C3a from C3. Subsequently, a C3a/C3 specific 3F7E2-immunoaffinity column was developed and apheresis of C3a/C3 and associates was performed. Finally, a proteomic analysis was carried out for identification of apheresis products. C3a/C3 was liberated from the 3F7E2-column together with 278 proteins. C3a/C3 interaction specificity was validated by using a haptoglobin immunoaffinity column as control and biostatistic analysis revealed 39 true C3a/C3 interactants.

**Conclusion:** A novel and functionally active mAb was developed against complement factor C3a/C3 and used in a specific immunoaffinity column that allows apheresis of C3a/C3 and associates and their identification by proteomic analysis. This methodological approach of developing specific antibodies that can be used as immunoaffinity reagents to design immunoaffinity columns for elimination and further identification of associated proteins could open new avenues for the development of tailored immunotherapy in various complement-mediated or autoimmune diseases.

## INTRODUCTION

Complement factor C3 is the central component of the complement cascade representing the junction point of the classical, alternative and lectin pathways and is split by the C3 convertase serine protease into C3a and C3b (*Ricklin et al., 2016*). It is present in high

Corresponding author
Wolfgang Winnicki,
wolfgang.winnicki@
meduniwien.ac.at

concentrations of up to 4 g/L in human blood and is mainly produced in the liver, but also other organs such as the kidneys are involved in its production (*Morgan & Gasque, 1997*). C3 is also known as acute phase protein and its high concentration in inflammatory conditions indicates biological functions beyond the usual contribution to the complement cascade activation. The N-terminal 77 amino acid split product C3a has a short half-life and is termed anaphylatoxin (*Haas & Van Strijp, 2007*; *Pasupuleti et al., 2007*) because of its mediatory function in various infections and inflammatory diseases (*Thurman et al., 2007*) via its specific C3a receptor (C3aR) (*Humbles et al., 2000*).

In the last years complement inhibition has become pivotal in the treatment of diseases such as paroxysmal nocturnal hemolysis (PNH) (*Dmytrijuk et al., 2008*; *Kim et al., 2010*), atypical hemolytic uremic syndrome (aHUS) (*Cofiell et al., 2015*; *Rathbone et al., 2013*), hereditary angioedema and generalized myasthenia gravis (*Ricklin et al., 2018*). Mutations in C3, among others, have been shown to be involved in the pathogenesis of aHUS (*Siomou et al., 2016*) and it has been elaborated in detail which mutations cause dysregulation of complement cascade activation (*Schramm et al., 2015*).

Recent reports in the literature show that C3 interacts with proteasome (*Lonnroth et al., 2016*), fibronectin (*Hautanen & Keski-Oja, 1983*), fibrinogen (*King et al., 2015*), activated platelets (*Hamad et al., 2010*), interleukin-2 (*Bartok et al., 1989*) and in total with at least 50 proteins (*Ricklin & Lambris, 2013*; *Sahu & Lambris, 2001*). This has aroused curiosity among scientists to identify further interaction partners as candidates for therapeutic modulation. Moreover, C3 itself has the potential to be a hot spot for complement-targeted therapeutic interventions (*Dobo, Kocsis & Gal, 2018*; *Liszewski et al., 2017*; *Mastellos et al., 2016*; *Salvadori, Rosso & Bertoni, 2015*; *Tatapudi & Montgomery, 2017*), and advances in the design of C3-targeted drug candidates as novel immunotherapeutics have been reported in recent years, including antibodies and peptides such as the compstatin analogs Cp40, APL-1, APL-2, APL-9, AMY-101 and AMY-103 (*Fremeaux-Bacchi & Legendre, 2015*; *Hawksworth et al., 2017*; *Katschke et al., 2009*; *Mastellos et al., 2015*; *Reddy, Siedlecki & Francis, 2017*; *Ricklin & Lambris, 2016*; *Ricklin et al., 2018*; *Subias Hidalgo et al., 2017*). The potential therapeutic applications and fields of research for these agents are broad and include C3 glomerulopathy, PNH, AB0 incompatible kidney transplantation, sepsis, age-related macular degeneration, choroidal neovascularization, warm autoimmune hemolytic anemia, chronic obstructive pulmonary disease and periodontal disease (*Ricklin et al., 2018*, *2016*). Furthermore, high concentrations of C3a/C3 in the circulation were identified under a variety of inflammatory conditions, including neurological diseases, cardiovascular diseases, asthma, cancer and transplant rejection (*Lines et al., 2016*; *Morgan, 2015*; *Pio, Corrales & Lambris, 2014*; *Ricklin et al., 2016*; *Sacks et al., 2013*; *Schmudde, Laumonnier & Kohl, 2013*).

The increasing knowledge of the central involvement of C3a/C3 in a variety of clinical aspects has encouraged us to develop a tool to reduce C3a/C3 from the circulation. Therefore, a novel monoclonal antibody (mAb) against complement factor C3 (3F7E2-mAb) was generated in a first step, which specifically binds to the anterior part of the alpha chain, the C3a region. In a second step functional tests of the 3F7E2-mAb were performed to investigate its inhibitory potential of C3 cleavage through Zymosan-induced

activation of the complement cascade in human serum. As third step, a novel C3a/C3-specific immunoaffinity column was developed using this mAb, which enables apheresis of the activation product C3a as well as the intact C3 and degraded forms and associates. In a fourth step, C3a/C3 associates, even beyond those already known, were identified by proteomic analysis. A haptoglobin-specific immunoaffinity column (3A8-mAb) (*Wagner et al., 1996*) was generated as control to enable by bioinformatic analysis exclusion of nonspecific binders to matrix components and antibody backbone.

## MATERIALS AND METHODS

### Preparation of the immunogen

Human serum diluted in PBS was injected into a size exclusion HPLC column (Ultraspherogel SEC 4000; Beckman Instruments, San Ramon, CA, USA) using PBS as mobile phase. The peak eluting at 5.16 min representing the void volume of the column was collected into four fractions (Fig. S1). These fractions contained the C3 protein and C3-associates together with other large serum proteins. These protein moieties contained in these fractions were pooled and concentrated using Centricon-30 (Amicon Co., Beverly, MA, USA) and further purified by re-chromatography.

### Antibody generation

A 4 week old Balb C mouse was immunized with human complement factor C3 (60 μg) purified from human serum by size exclusion chromatography as described above. Following three immunizations in a 2 weeks interval spleen cells were fuzed with the hypoxanthine–aminopterin–thymidine sensitive murine myeloma cell line 63A8.536 using the poly ethylene glycol method as described earlier (*Wagner et al., 1996*). Outgrowing clones were screened by a multi-slot immunoblotting device (Millipore, Bedford, MA, USA) (Fig. S2). Clones recognizing SDS denatured C3 under non-reducing conditions were kept, sub-cloned and further used for experiments. Clone number 18 as depicted in Fig. S2 was selected for its capability in immune purification of the C3 molecule. This antibody was finally named 3F7E2.

### Purification of 3F7E2-antibody

The well producing antibody clone 3F7E2 was grown for 72 h in tissue culture flasks using RPMI1640 supplemented with 10% fetal calf serum containing penicillin/streptomycin as culture medium. The incubation conditions at a humidified (95% $H_2O$) incubator were 5% $CO_2$ and 37 °C. The tissue culture supernatant was filtered through a 0.22 μm filter. The resultant tissue culture fluid was loaded onto a HiTrap protein G column (GE Healthcare, Uppsala, Sweden) for purification of the murine IgG1 mAb. The purified mAb was then dialyzed against phosphate-buffered saline (PBS) at 4 °C and then further used for preparing an immunoaffinity column.

### Gel electrophoresis and immunoblotting

One μL of human serum was diluted in near neutral pH sample buffer and loaded onto a PAGE gel (10%) and run using a Tris Glycine buffer omitting SDS for native gel and

including SDS for C3a/C3 protein characterization. For detection of C3a 12% Mini-Protean TGX™ gels (456-1046; BioRad, Munich, Germany) and a Tris-Tricine SDS running buffer (161-0744; BioRad, Munich, Germany) were used. Native C3a was obtained from Complement Technology Inc. (Tyler, TX, USA), which had been purified from human donors, whereby its natural structure had been preserved.

Gels were blotted onto nitrocellulose using a semidry blotting device. The blotted nitrocellulose membrane was blocked with BM Chemiluminescence Blotting Substrate (POD) (Roche Molecular Biochemicals, Mannheim, Germany) and then developed using 3F7E2-mAb by incubating over night at 4 °C. Following two washes in Tween–phosphate buffered saline (TPBS) the blot was further incubated with POX conjugated goat anti mouse Ab 1:10,000 (PO447, DAKO, Denmark) for 1 h at room temperature and further developed using the chemiluminescence method with BM Chemiluminescence Blotting Substrate (POD) (Roche Molecular Biochemicals, Mannheim, Germany) and recording the site of antibody binding with luminescence imaging. Images were recorded using Fusion 3 software at the image analyzer unit (Fusion FX, Vilber Lourmat, Eberhardzell, Germany). C3 activation experiments were carried out by incubation of serum at 37 °C for various time periods. Obtained C3 split products were identified by immunoblotting.

## Functional study using Zymosan activation

Ten μL of human serum were incubated in polypropylene reaction tubes of 0.5 mL (Sarstedt, Nuembrecht, Germany) with 5 μg and 10 μg of activated Zymosan for 15 min at 37 °C. As controls the antibody alone or zymosan suspension buffer alone were included in the test. The activation process was stopped by adding 1 μL of the experimental volume into sample buffer containing EDTA, which was heated for 3 min at 95 °C and loaded onto an SDS Page Gel (12% Mini-Protean TGX™ gel, 456–1046; BioRad, Munich, Germany) under nonreducing conditions. The 12% gel was run using Tris-Tricine SDS buffer (BioRad, Munich, Germany) and the resultant gel was transferred to nitrocellulose by semidry blotting and further developed using either mAb 3F7E2 or ab48342, rabbit polyclonal antibody to C3 (Abcam, Cambridge, UK) as described above.

## C3 ELISA

Goat anti mouse precoated plates (Reacti-Bind™ Antibody Coated Plates, Pierce Biotechnology, Rockford, IL, USA) were washed once with PBS and the 3F7E2-mAb was bound onto the plate by incubating hybridoma cell supernatant onto the plate for 2 h. After one wash with TPBS 100 μL of samples were loaded together with a C3 standard series. Following an incubation period of 2 h the ELISA plate was washed three times with TPBS using an ELISA washing machine and the rabbit anti human C3 Ab conjugated with biotin (ab48342; Abcam, Cambridge, UK) diluted 1:2,500 using Assay Diluent (EL-ITEME2, RayBiotech Inc., Norcross, GA, USA) was incubated for 1.5 h under constant shaking. Following three washes with the ELISA washing machine with TPBS the streptavidin HRP was bound to the biotin while incubating in Assay Diluent (EL-ITEME2, RayBiotech Inc., Norcross, GA, USA) for 30 min at room temperature under constant shaking. Following three washes with TPBS the TMB substrate chromogen solution (Peroxidase Substrate

Solution, KPL, Gaithersburg, MD, USA) was added and kept in dark for 10 min. The reaction was stopped with 1 M HCl and read with the ELISA reader (405 nm). Each value was calculated according the standard curve and samples were measured in quadruplicate.

## Production of a 3F7E2-immunoaffinity column

Protein G column purified IgG 1 mAb was ligated to cyanogen bromide Sepharose for immunoaffinity purification of human factor C3a/C3. The 3F7E2-sepharose column was washed with wash buffer 1 (50 mM Tris-HCl, 0.5 M NaCl, 0.1% Triton X-100, pH 8.0) and wash buffer 2 (50 mM Tris-HCl, 0.5 M NaCl, 0.1% Triton X-100, pH 9.0) and finally with triethanolamine (TEA) elution buffer (50 mM TEA, pH 11.3). Following the preparation procedure of the immunoaffinity column human precleared serum was loaded onto the 3F7E2-mAb column by recirculation at 4 °C for 2 h. The loaded column was then washed with 10 column volumes of wash buffer (0.01 M Tris-HC1, pH 8.0, at 4 °C, 0.14 M NaC1, 0.025% NaN$_3$, 0.5% Triton X-100 and 0.5% sodium deoxycholate), 10 column volumes of wash buffer 1 followed by wash buffer 2. Following the washing procedure, the protein was eluted from the 3F7E2-column using TEA elution buffer, pH 11.3 into 1 M Tris-HCl, pH 6.7. Purified C3a/C3 was loaded onto 10% SDS PAGE gels and stained with Coomassie Brilliant Blue followed by destaining. Destained gels were incubated in glycerol H$_2$O for 1 h. Individually stained bands were cut out and subjected to proteomic analysis.

A haptoglobin immunoaffinity column (3A8-mAb) was prepared using the same method as described earlier (*Wagner et al., 1996*). Protein affinity purification and electrophoresis was performed using the same method as described for C3 affinity purification.

## Protein gel electrophoresis for proteomic analysis

Equal amount of protein originating from haptoglobin and C3 immunoaffinity purification, respectively, were loaded onto separate lanes and different SDS PAGE gels both run under reducing conditions. The gels were stained by Coomassie Brilliant Blue followed by destaining. From both the C3 and haptoglobin containing lanes the same sized slice was cut out covering the entire molecular weight range and submitted for proteomic analysis.

## Mass spectrometry
### NanoLC-MS analysis
The methodology used has already been described in earlier studies (*Qiao et al., 2016*; *Wissel et al., 2016*; *Zess et al., 2018*). In brief, the UltiMate 3000 RSLC nano system (Thermo Fisher Scientific, Amsterdam, Netherlands) was combined with Q Exactive HF mass spectrometer (Thermo Fisher Scientific, Bremen, Germany), using a Proxeon nanospray source (Thermo Fisher Scientific, Odense, Denmark). As trap and analytical columns PepMap C18 columns (Thermo Fisher Scientific, Amsterdam, Netherlands) were used. Peptide elution was performed at a gradient as previously reported using water and

formic acid as well as water, acetonitrile and formic acid in the mobile phase. Exact technical settings are described in our previous work (*Reiter et al., 2019*).

### Data processing protocol

A detailed description is available in previous articles (*Qiao et al., 2016*; *Wissel et al., 2016*; *Zess et al., 2018*).

In short, Proteome Discoverer (version 2.1.0.81; Thermo Scientific, Bremen, Germany) was fed with the raw files in order to identify peptides. As search engine MSAmanda v2.0.0.9849 (*Dorfer et al., 2014*) was used connected with SwissProt human database (20169 sequences; 11315794 residues). Exact parameter settings are indicated in previous work (*Zess et al., 2018*). The tool ptmRS was used to investigate the locus of the posttranslational peptide modification sites (*Taus et al., 2011*). The IMP apQuant tool was used for peptide and protein quantification (*Doblmann et al., 2019*). The software calculates a "protein area" from the sum of all the peptide areas assigned to each identified protein. Three technical replicate measurements were performed for the C3-affinity sample, and likewise for the haptoglobin-affinity sample. Enrichment in the C3 sample was calculated as log2 of the geometric mean of the three area ratios for each protein. A $p$-value was calculated from a 2-sided $t$-test for a comparison of the protein areas in the three C3 replicates and the three haptoglobin replicates. Trypsin and keratins were removed from the list as likely contaminants. Immunoglobulin fragments were also eliminated from calculations. Strong and consistent (10-fold, $p < 0.05$, respectively) presence of a protein in the C3-bait replicates as compared to the haptoglobin-bait replicates was considered C3a/C3-specific binding. These criteria were set as cut off at the calculation method.

## Ethics approval and consent to participate

All experiments were performed in accordance to the Helsinki Declaration and approved by the Ethics Committee of the Medical University of Vienna (EK721/2007). Serum samples were collected from adult subjects older than 18 years of age who gave written informed consent. All methods have been performed in accordance with the current regulations and guidelines. Animals were kept at the Center for Biomedical Research at the Medical University of Vienna. The Austrian Federal Ministry of Education, Science and Research provided full approval for this research (BMWF66.009/0046-II) in accordance to the European Convention for the Protection of Vertebrate Animals used for scientific purposes.

## RESULTS

### Immunoblotting of C3a/C3 and antibody specificity

By testing the 3F7E2-mAb for its specific recognition site at the C3a/C3 molecule, it was demonstrated that there is no binding under reducing conditions (Fig. 1B). Immunoblotting recognized C3 at a molecular weight of 190 kDa and more specifically, commercial C3a obtained from Complement Technology Inc. (Tyler, TX, USA) at a molecular size of approx. 9 kDa when the PAGE gel was run under non-reducing conditions (Fig. 1A). A conformational structure together with a specific amino acid

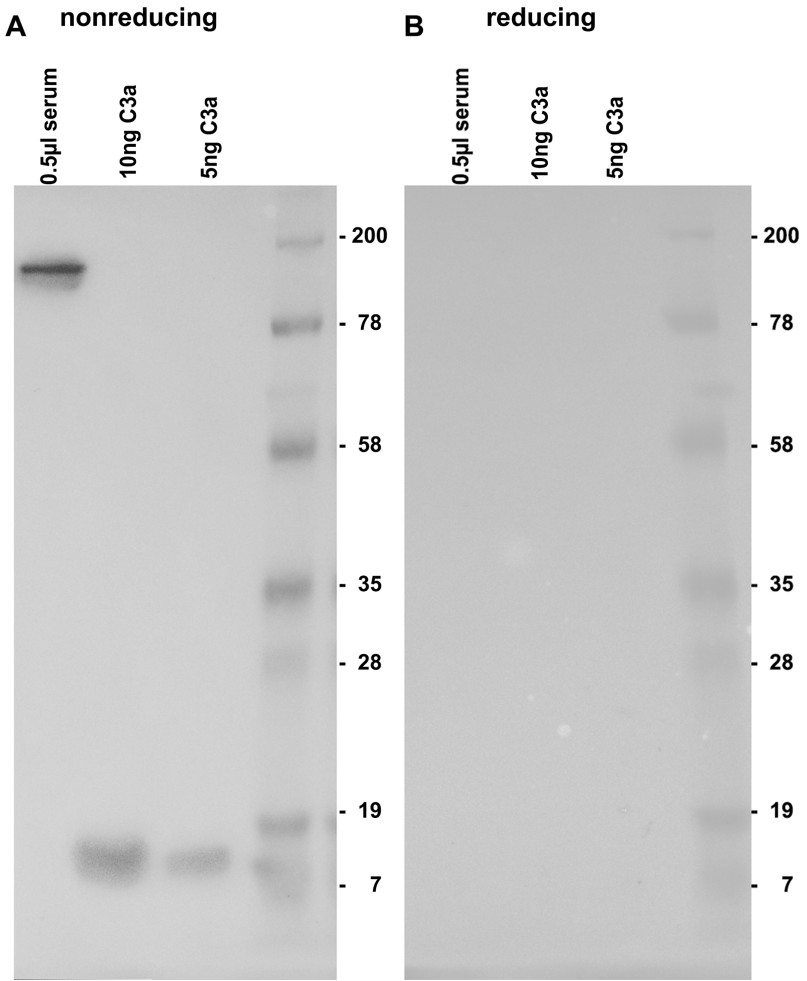

**Figure 1 Immunoblotting using 3F7E2 of human serum and commercial C3a.** Five hundred nanoliter (nL) of human serum loaded onto lane 1, 10 ng purified C3a loaded onto lane 2, 5 ng purified C3a loaded onto lane 3 and a molecular weight marker loaded onto lane 4 were run on a 12% SDS PAGE gel under non-reducing (A) and under reducing (B) conditions. The blotted membrane was blocked and incubated with 3F7E2-mAb over night at 4 °C. The primary mAb was followed by a goat anti mouse POX conjugated Ab 1:10,000 and incubated for 60 min at room temperature and finally developed using BM Chemiluminescence Blotting Substrate (POD), Roche Molecular Biochemicals, Mannheim, Germany, for picture recording. It shows two grouped gels divided by a white space representing one representative experiment out of three.             

sequence located at the anterior part within the C3a molecule was revealed to represent the antibody binding site.

## Native gel electrophoresis

Initial size exclusion chromatography experiments for preparing the immunogen (Fig. S1) led to the hypothesis that C3 is present in association with other proteins under native conditions. In order to analyze the hydrodynamic size of the molecule and verify that the native C3a/C3 protein is associated in human serum with further proteins we determined its mobility in native gel. This revealed a spreading of C3 positive bands over a broad range in each of nine tested individuals and the pattern was unique to each individual.

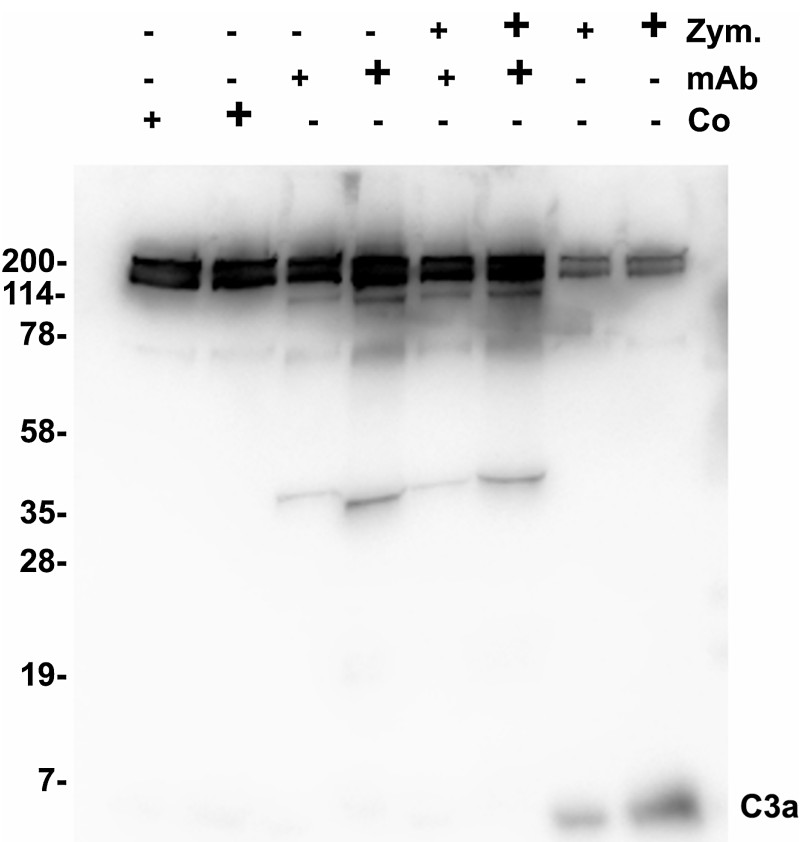

**Figure 2 Zymosan activation of C3.** One μL of Zymosan activated serum was loaded onto each lane of a 12% SDS PAGE which was run under nonreducing conditions using Tris-Tricine as running buffer. Following semidry blotting of the gel the nitrocellulose membrane was incubated with 3F7E2-mAb after incubation in blocking buffer. The antibody binding was developed with POX conjugated goat anti mouse (1:10,000). For binding visualization BM Chemiluminescence Blotting Substrate (POD) (Roche Molecular Biochemicals, Mannheim, Germany) was used which was recorded with luminescence imaging. Human serum was incubated with control buffer in lanes 1 (5 μL) and 2 (10 μL), with 3F7E2-mAb 2.5 μg (+) and 5 μg (+) in lanes 3 and 4, with Zymosan 5 μg (+) and 10 μg (+) together with 3F7E2-mAb 2.5 μg (+) and 5 μg (+) in lanes 5 and 6 and with Zymosan alone at 5 μg (+) and 10 μg (+) in lanes 7 and 8. The band at 43 kDa in lanes 3–6 represents one part of mAb-3F7E2. This blot illustrates one representative experiment out of three. Abbreviations: Co, control; mAb, monoclonal antibody; Zym, Zymosan.                

Nevertheless, the main dominant band in all samples was at a size of 190 kDa, which complies with the expected size of native C3 (Fig. S3).

## Functional analysis of 3F7E2-mAb

Functional tests were performed to show a possible biological activity of 3F7E2-mAb in the activation process of the complement cascade. The activation process of human serum was initiated with activated Zymosan at 37 °C. An inhibitory potential of more than 70% (assessed by densitometric scanning) was achieved when purified 3F7E2-mAb was added before addition of Zymosan and the incubation period was restricted to 15 min (Fig. 2). Similar findings could be obtained by ELISA testing after 15 min Zymosan activation,

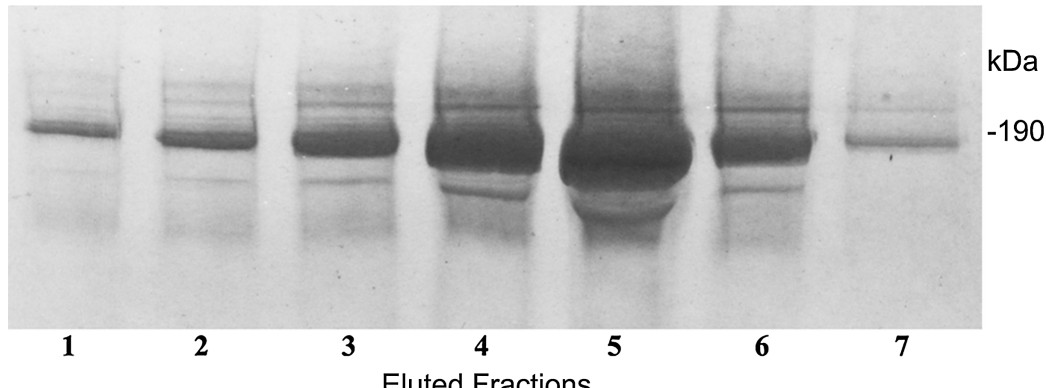

**Figure 3 Immunoapheresis elution profile of C3.** Five mL human serum was recirculated over the 3F7E2-column for 3 h at 4 °C. Following the column washing procedure, the bound antigen was eluted by TEA buffer with 10 column volumes. C3 was liberated from the column as a complex moiety and collected in seven individual fractions. Maximal protein liberation from the column was shown in elution fraction 5. An aliquot of each eluted fraction was loaded onto an SDS PAGE gel and run under non-reducing conditions. Proteins were stained by Coomassie Brilliant Blue. This experiment was repeated three times.

which showed an inhibition of C3 consumption by 70% via addition of 3F7E2-mAb (490.1 ng/mL ± 20.1 ng/mL vs. 150.3 ng/mL ± 9.7 ng/mL).

## Immunoapheresis of C3a/C3 and identification of bands

In a further technique to define the complex molecular structure, we sought for a way of immunopurification of these C3a/C3 compounds. For this purpose we generated an immunoaffinity 3F7E2-chromatography column for immunoapheresis of native C3a/C3 from human serum. Following the selection of the mAb 3F7E2 and preparation of the column, the C3a and C3 moiety (Fig. 3) could be purified under native conditions. The eluted fractions using TEA buffer were loaded onto an SDS-PAGE gel and electrophoresed under non-reducing conditions.

After SDS PAGE gel electrophoresis of eluted C3 fractions, staining of the gel using Coomassie Brilliant Blue revealed the main C3 band at 190 kDa and C3a as well as a C3 degradation product band at 9 and 8 kDa, respectively (Fig. 4).

## Proteomic identification of C3a/C3 associates

Narrowly cut sections of the 9 kDa and 8 kDa bands were submitted to proteomic analysis. The resultant peptide finger print revealed peptides matching to amino acids 672–748 complying with the primary structure of full length C3a. The peptide fingerprint of the marginally smaller band (8 kDa) was also matching with the C3a sequence however missing a fragment from C-terminal half (Table 1) and most likely represents a degradation product. The peptide finger print of the 190 kDa band provided full length C3 sequence peptides (Fig. 4).

In the proteomic analysis a number of proteins or their fragmentation products were co-purified. Out of 278 proteomically identified proteins, 39 fulfilled the cut off criteria for specifically binding to the C3a/C3 molecule obtained by immunoapheresis.

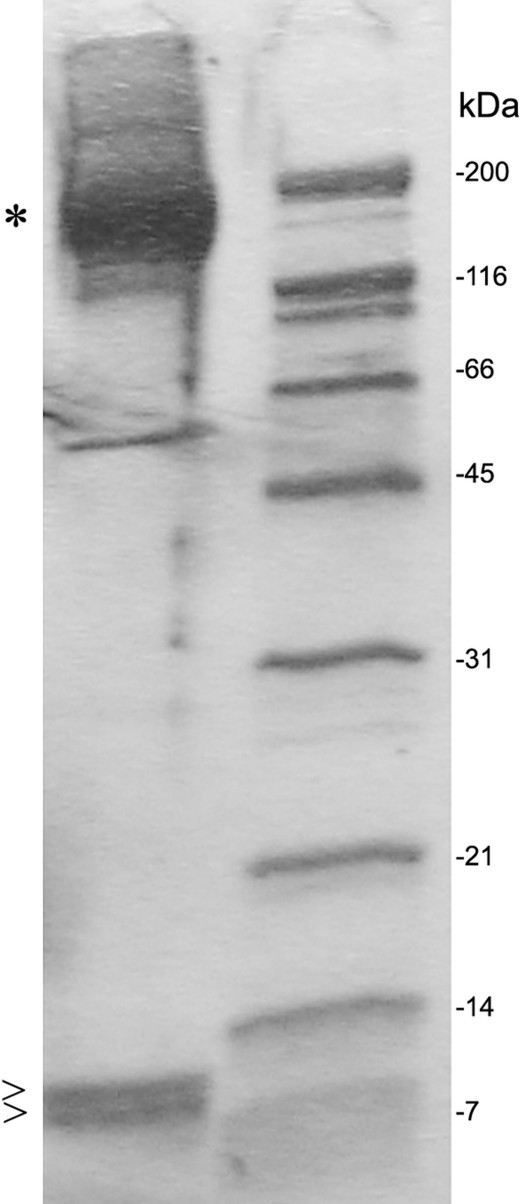

**Figure 4 SDS PAGE gel electrophoresis of eluted fractions and preparation for proteomic analysis.** Individual fractions were loaded onto an SDS PAGE gel and bands at 190 kDa (*), were identified as C3, and bands at 9 and 8 kDa (>) were identified as C3a or C3 degradation product by selective proteomic analysis. The C3 degradation product band at 8 kDa missed a fragment from C-terminal half of C3a at the proteomic analysis. A molecular size marker is represented at the right lane.

This cut off criteria were defined as proteins with strong (>10-fold) and consistent ($p$-value < 0.05) regulation in three technical replicates, consistent with C3a/C3-specific interaction. As depicted in Fig. 5 these 39 proteins, also listed in Table 2, had an at least 10-fold higher quantity of binding to C3a/C3 as compared to the haptoglobin control column.

**Table 1  Identified peptides and individual peak size area of bands at 8 kDa and 9 kDa.** Amino acid position numbering according to unprocessed protein C3 is presented in column one and peptide sequence in column two. Individual peak size area of peptides composing C3 degradation product derived from band at 8 kDa is given in column three and peak size area of peptides composing C3a derived from band at 9 kDa is shown in column four. The virtual peptide size (Theo. MH+ [Da]) is given in column five. The peptide fingerprint of the smaller band at 8 kDa was matching with the C3a sequence missing a fragment from C-terminal half.

| Position in protein | Sequence in protein | ApQuant area | | Theo. MH+ [Da] |
|---|---|---|---|---|
| | | 8 kDa | 9 kDa | |
| [672–679] | R.SVQLTEKR.M | 1,53E+08 | 2,19E+08 | 961.5 |
| [680–688] | R.MDKVGKYPK.E | 9,11E+07 | 8,36E+07 | 1,081.6 |
| [692–704] | R.KCCEDGMRENPMR.F | 1,45E+08 | 2,06E+08 | 1,692.6 |
| [711–722] | R.TRFISLGEACKK.V | 4,79E+07 | 9,78E+07 | 1,398.7 |
| [713–722] | R.FISLGEACKK.V | 2,60E+09 | 4,59E+09 | 1,141.6 |
| [722–735] | K.KVFLDCCNYITELR.R | | 4,09E+07 | 1,808.8 |
| [741–748] | R.ASHLGLAR.S | | 2,08E+08 | 824.5 |

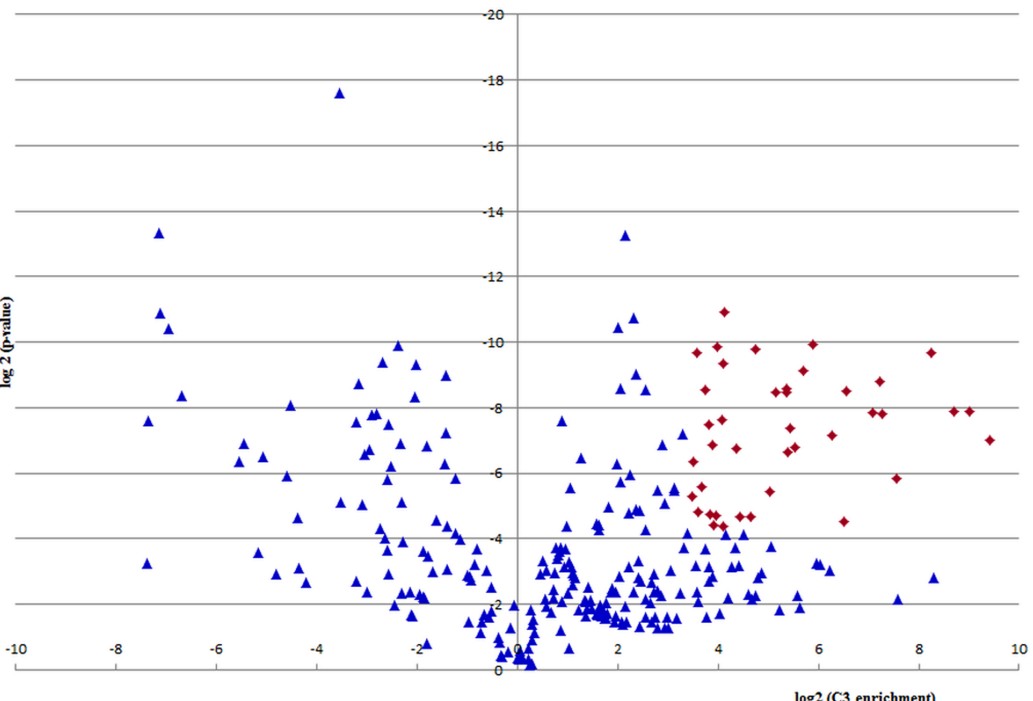

**Figure 5  Quantification of C3a/C3-binding.** A volcano plot was derived by depicting log2 enrichment in the C3a/C3 sample on the x-axis, and log2 of the p-value on the y-axis. Proteins with strong (>10-fold) and consistent (p-value < 0.05) regulation in three technical replicates are considered specific C3a/C3 interactors and are depicted as red diamonds.

In addition to some ubiquitous proteins and well-known C3 interaction partners, some particularly interesting associated proteins that may represent potential therapeutic targets were identified. Among the detected proteins were several biologically important factors, such as: (1) regulators of the complement cascade: Complement factor H, Complement factor H-related proteins 1 and 2, Complement C1q, Complement C4-B and C4b-binding protein

**Table 2 Proteomic analysis and peptide identification of C3a/C3 associates.** Column four marks established C3 partners known from previous literature.

| Protein name | Log2 C3 enrichment | Log2 (*p*-value) | Established C3 associates |
|---|---|---|---|
| Inter-alpha-trypsin inhibitor heavy chain H2 | 9.410 | −6.996 | |
| Complement C1q subcomponent subunit A | 9.001 | −7.875 | x |
| Protein disulfide-isomerase | 8.703 | −7.868 | |
| Inter-alpha-trypsin inhibitor heavy chain H1 | 7.567 | −5.830 | x |
| Histidine-rich glycoprotein | 7.271 | −7.791 | |
| Galectin-3-binding protein | 7.220 | −8.813 | |
| Kininogen-1 | 7.077 | −7.837 | |
| Complement component C8 alpha chain | 6.551 | −8.495 | |
| Complement factor H-related protein 1 | 6.518 | −4.503 | x |
| Complement C1q subcomponent subunit C | 6.263 | −7.156 | x |
| Inter-alpha-trypsin inhibitor heavy chain H4 | 5.880 | −9.936 | |
| Complement component C8 beta chain | 5.710 | −9.139 | x |
| Protein AMBP | 5.527 | −6.775 | |
| Apolipoprotein B-100 | 5.433 | −7.360 | |
| C4b-binding protein alpha chain | 5.391 | −6.635 | x |
| Serum albumin | 5.375 | −8.469 | x |
| Alpha-2-macroglobulin | 5.354 | −8.576 | |
| Serotransferrin | 5.154 | −8.479 | x |
| Apolipoprotein A-I | 5.025 | −5.442 | x |
| Prolactin-inducible protein | 4.734 | −9.781 | |
| Complement factor H | 4.643 | −4.658 | x |
| Complement component C8 gamma chain | 4.434 | −4.670 | x |
| Vitamin K-dependent protein S | 4.356 | −6.739 | |
| Arachidonate 12-lipoxygenase, 12R-type | 4.127 | −10.923 | |
| Ficolin-3 | 4.096 | −9.335 | |
| Complement C4-B | 4.092 | −4.391 | x |
| Complement factor H-related protein 2 | 4.088 | −7.610 | |
| Ceruloplasmin | 3.976 | −9.843 | |
| Hornerin | 3.953 | −4.689 | |
| Thrombospondin-1 | 3.912 | −4.416 | |
| Histone H2AX | 3.877 | −6.872 | |
| Actin, alpha cardiac muscle 1 | 3.836 | −4.724 | |
| Kallikrein-5 | 3.822 | −7.483 | |
| Skin-specific protein 32 | 3.733 | −8.531 | |
| Alpha-1-antitrypsin | 3.662 | −5.592 | |
| HLA class I histocompatibility antigen, A-68 alpha chain | 3.597 | −4.802 | |
| Short-chain dehydrogenase/reductase family 9C member 7 | 3.571 | −9.662 | |
| Corneodesmosin | 3.516 | −6.361 | |
| Serpin A12 | 3.484 | −5.280 | |

and Complement component C8; (2) serine protease inhibitors: Serpin A12, Inter-alpha-trypsin inhibitor and Protein AMBP as well as (3) intracellular proteins such as Histone H2AX.

Further potential C3-interacting partners, with proteins below selected cutoff criteria and enrichment value above 0, located in the right half of the volcano plot of Fig. 5 are listed in Table S1.

## DISCUSSION

The field of complement-targeted drug research has fundamentally changed over the last decades (*Sim, Schwaeble & Fujita, 2016*). The complement system offers a number of potential drug targets, most of which are readily and abundantly available in the circulation or on cell surfaces (*Ricklin & Lambris, 2007*). However, interference with several distinctive functions of the complement system requires tailored drug design approaches, as complement is driven by protein–protein interactions and conformational conversions (*Gros, Milder & Janssen, 2008*; *Ricklin & Lambris, 2007*; *Yaseen et al., 2017*). The development of drugs based on engineered proteins, antibodies and peptides (*Katschke et al., 2009*; *Lindorfer et al., 2010*) has led to relevant discoveries of complement-targeted drugs (*Ricklin et al., 2018*).

There are currently five therapeutics targeted against complement factors on the market, which are approved for four indications (hereditary angioedema, PNH (*Rother et al., 2007*), aHUS (*Fakhouri & Fremeaux-Bacchi, 2013*) and generalized myasthenia gravis (*Ricklin et al., 2018*)) and numerous drug candidates have been (*DeZern et al., 2014*; *Risitano et al., 2014*) and are being evaluated in clinical and in vitro trials targeting different levels (*Schmidt et al., 2013*) of the complement cascade.

The therapeutic success of Eculizumab as mAb against the complement component C5 has certainly been the driving force behind complement-targeted drug research in recent years. However, there is evidence that the effect of anti-C5 therapy may be limited due to strong involvement of C3 activation in particular circumstances and treatment strategies that target upstream of C5 activation may therefore be beneficial in selected indications (*Bomback et al., 2012*; *Nester & Smith, 2016*; *Ricklin et al., 2018*).

In this study a novel mAb was developed against the critical C3a activation split product of complement factor C3 (3F7E2-mAb) and was used in a C3a/C3 specific immunoaffinity column that allows to eliminate C3a and C3 combined with binding partners by means of apheresis. In addition, a previously unknown C3 degradation product could be identified. This represents the 8 kDa band shown in Fig. 4, which was identified in proteomic analysis as peptides spanning over the anterior part of C3a, but missing a fragment from the C-terminal half which was not described in the literature before. This study emphasizes that the protein C3 and its activation product C3a is not circulating under native conditions as a single molecule, it rather associates with other factors, which some of them represent regulators of the complement cascade.

It has been demonstrated in animal models that C3a is centrally involved in disease processes such as airway disease inflammation and progression of glomerulonephritis to glomerulosclerosis (*Morigi et al., 2016*). This is mediated through the ubiquitously expressed C3aR. It has been further shown in previous studies that elimination of C3 and

the complement receptor CR3 in murine glial cells leads to an accumulation of amyloid plaques in mice as the C3/CR3 system contributes to elimination of misfolded protein fragments in the brain, thus absence of C3 or CR3 has negative effects on fibrillary amyloid beta clearance by microglia (*Fu et al., 2012*). Consequently, it is conceivable that in peripheral blood this protein acts as scavenger protein to eliminate protein remnants from intracellular origin. Intracellular proteins such as histones are present at sites of inflammation, trauma and cell death and, if left unaffiliated, can cause injuries at the glomerular capillary wall (*Kumar et al., 2015*; *Nakazawa et al., 2017*). Considering the tight margin between inhibition, physiological state and over-activation of the complement cascade, elimination of elevated concentrations of C3a and C3 including interaction partners has potentially far-reaching therapeutic effects.

In this study a biologically relevant inhibitory activity of the 3F7E2-mAb was confirmed by Zymosan activation experiments of the complement cascade. The 3F7E2 antibody inhibited C3a generation over 70% suggesting that the antibody is potent in inhibiting the conversion of C3–C3a and C3b under described conditions. In this context, it is important to consider the biological impact of therapeutic inhibition of C3. Patients with C3 inhibition may have an increased susceptibility to infections and should possibly be immunized against bacterial infections, however there is evidence that a low C3 concentration of less than 20% of the normal range in the circulation is sufficient to maintain adequate activation of complement response in adults with fully developed adaptive immunity (*Da Silva et al., 2016*; *Ricklin et al., 2016*).

In our study protease inhibitors such as Serpin A12 were identified as associate proteins of C3a/C3, which might either act as veiling factors or represent active inhibitors of complement convertases and thereby modify the extent of complement activation. Further proteins with proteinase inhibitory function were detected, such as Protein AMBP and the Inter-alpha-trypsin inhibitor, a representative of a glycosaminoglycan-protein complex with inhibitory activity against trypsin, chymotrypsin, neutrophil elastase and plasmin (*Potempa et al., 1989*; *Zhuo, Salustri & Kimata, 2002*). The binding of protease inhibitors most likely to the anterior part of the C3 alpha chain (C3a region) or various factors of different steps of the complement cascade such as Complement factor H, Complement factor H-related proteins 1 and 2, Complement C1q, Complement C4-B and C4b-binding protein and Complement component C8 reflects the tightly regulated equilibrium in which C3 is central. It provides the constant immune surveillance of tissue under steady state conditions. However, this is susceptible to changes depending on various stimulators acting on C3 and associates. For example, infections can divert the complement-mediated immune surveillance towards over-activation and thus tissue destruction, which can lead to nephritis and other autoimmune diseases. A number of protease inhibitors identified as C3a/C3 associates have not yet been considered as regulators of the complement cascade.

Another important observation of this study is that highly noxious extracellular histones as potential causative agents for glomerular nephritis and acute kidney injury (*Kumar et al., 2015*; *Nakazawa et al., 2017*) are bound to C3a/C3. This is consistent with the fact that the complement system has some "waste disposal" capacity. In this line of

thought, it might be speculated that the positively charged histones are inactivated by C3 and deposited for degradation in macrophages. C3a/C3 might bind to misfolded or partly degraded proteins and transports them to specific receptors on macrophages or glial cells where they become degraded (*Fu et al., 2012*).

## CONCLUSION

A novel functionally active mAb (3F7E2-mAb) was developed against complement factor C3a/C3 and used in a specific immunoaffinity column that enables apheresis of C3a/C3, their degradation products and associates. In future, this unique methodological approach could be tailored for specific immunoapheresis in various complement-mediated or autoimmune diseases.

### Funding

The authors received no funding for this work.

### Competing Interests
The authors declare that they have no competing interests.

### Author Contributions
- Wolfgang Winnicki conceived and designed the experiments, analyzed the data, prepared figures and/or tables, authored or reviewed drafts of the paper, approved the final draft.
- Peter Pichler performed the experiments, analyzed the data, prepared figures and/or tables, authored or reviewed drafts of the paper, approved the final draft.
- Karl Mechtler performed the experiments, contributed reagents/materials/analysis tools, authored or reviewed drafts of the paper, approved the final draft.
- Richard Imre performed the experiments, contributed reagents/materials/analysis tools, authored or reviewed drafts of the paper, approved the final draft.
- Ines Steinmacher performed the experiments, contributed reagents/materials/analysis tools, authored or reviewed drafts of the paper, approved the final draft.
- Gürkan Sengölge analyzed the data, authored or reviewed drafts of the paper, approved the final draft.
- Daniela Knafl performed the experiments, authored or reviewed drafts of the paper, approved the final draft.
- Georg Beilhack conceived and designed the experiments, contributed reagents/materials/analysis tools, authored or reviewed drafts of the paper, approved the final draft.
- Ludwig Wagner conceived and designed the experiments, performed the experiments, analyzed the data, contributed reagents/materials/analysis tools, prepared figures and/or tables, authored or reviewed drafts of the paper, approved the final draft.

## Human Ethics

The following information was supplied relating to ethical approvals (i.e., approving body and any reference numbers):

All experimental protocols were performed in accordance to the Helsinki Declaration and approved by the ethics Committee of the Medical University of Vienna (EK721/2007).

## Animal Ethics

The following information was supplied relating to ethical approvals (i.e., approving body and any reference numbers):

Animals were kept at the Center for Biomedical Research at the Medical University of Vienna. The Austrian Federal Ministry of Education, Science and Research provided full approval for this research (BMWF66.009/0046-II) in accordance to the European Convention for the Protection of Vertebrate Animals used for scientific purposes.

## Data Availability

The mass spectrometry proteomics data is available at the PRIDE Archive: PXD009829.

## Supplemental Information

Supplemental information for this article can be found online at http://dx.doi.org/10.7717/peerj.8218#supplemental-information.

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
