# Peer review of "A novel approach to immunoapheresis of C3a/C3 and proteomic identification of associates"

_PeerJ, doi:10.7717/peerj.8218_

## Round 0.1 · original submission · Major Revisions

Your manuscript has been reviewed by our external reviewers and found it suitable for publication in the Peer J, if you appropriately revise it.
I have read your study with my interest in this C3.

Looking forward to receiving your revision.

Thank you

Sincerely,
Cheorl-Ho Kim

Reviewer 1 ·

Basic reporting

I don’t consider myself to be qualified enough to estimate the quality of English in the text of the manuscript.

Experimental design

No comment

Validity of the findings

Except for the points for revision listed below, all experimental data are presented quite clear and properly discussed, raw data are supplied, conclusions are well stated.

Points for revision:

1) For me, the most intriguing result of the work presented in the manuscript is the data on the identity of low molecular weight antigens for 3F7E2-mAb from blood serum revealed after the immunoapheresis procedure. In the manuscript, these components are designated as ‘C3a’ and ‘C3a missing a C-terminal part’ (line 275, 282-283; descriptions for Figure 4 and Table 1); the second one is discussed as presumable ‘C3a degradation product’ (line 283). However, taking that there was no mistake in Table 1 preparation, both of these components could not be referred to as C3a. The C3a molecule has an arginine 748 on its C-terminus (or alanine 747 in the case of C3a des-Arg), while both revealed components have an extension 749-764 according to data presented in Table 1. In my opinion, the smaller component is better to describe as ‘missing a fragment/piece from C-terminal half’ but not as ‘missing a C-terminal part’. Both these components should be referred to as ‘C3 degradation products’. To my knowledge, the presence of such C3-derived species in human serum was not reported previously, and it should be discussed in the ‘Discussion’ section.

2) The data of the proteomic research of C3-interacting proteins presented in the manuscript are undoubtedly of high fundamental importance. That is why they should be described in more detail.
I recommend outlining in Table 2 which of the C3 partners were known previously and which were revealed for the first time.
To distinguish between specific and nonspecific interactions, the authors introduced cut-off criteria, which allowed to describe 39 proteins as specific C3-interacting partners among 278 ones identified totally. It seems worthy to list the remaining 239 proteins (or at least those located in the right half of the plot in Figure 5) in a separate supplementary file.

3) In this work, 3F7E2-mAb was demonstrated to have a complement inhibition activity. It is stated in the manuscript that ‘inhibitory potential of 3F7E2-mAb is shown to be dose-dependent’ (lines 260-261; description for Figure 2). In my opinion, the presented data are not enough to conclude a dose dependency. Only two concentrations of the antibody were tested, and they both revealed identical near-complete inhibitory effects according to Figure 2. Therefore, these results should be discussed more carefully or additional data confirming dose dependency should be presented.

4) Minor editing
Perhaps, it would be better to spell down some abbreviations, which are not quite commonly used, e.g. TPBS (probably, Tween-PBS; line 133), TEA (probably, triethanolamine; line 174).
It seems that ‘3’ is missed in the sodium azide formula (line 178).
From the description for Figure 2, it seems that lanes 1 and 2 represent identical samples, both loaded on the gel in the same volume. However, these lanes are accompanied by ‘+’ signs of different size. Thus, it should be corrected (or description for the Figure should be clarified).

Additional comments

This is an interesting and definitely significant study devoted to the development of a novel approach to the inhibition of complement activation, which is a serious medical task. The 3F7E2-mAb antibody seems to be promising for clinical applications aimed at limiting excessive complement activity both as a direct complement inhibitor, and as an immunoaffinity reagent for the apheresis procedure

·

Basic reporting

Abstract needs to be structural. Please refer to authors’ instructions on https://peerj.com/about/author-instructions/ for details.

Abbreviations should be written in their complete form when they are mentioned for the first time in the text. e.g. C3aR in line 52, IL-2 in line 62, TPBS in line 133, nL in the 1st line of figure 1’s legend.

Names of providers and manufacturers of consumables and instruments should be stated in parenthesis. e. g. “Roche Molecular Biochemicals, Mannheim, Germany” in different places throughout the text, “BioRad, Germany” in line 149, “Abcam, Cambridge, UK” in line 150.

In line 166, “und” should be changed to “and”.

Experimental design

Tissue culture (line 117) needs to be explained in more details including its culture media, timing, temperatures and etc.

Validity of the findings

No comment.

---

## Round 0.2 · accepted · Accept

The C3a/C3-related results will be appreciated

Reviewer 1 ·

Basic reporting

No comment

Experimental design

No comment

Validity of the findings

In the revised manuscript, all the questions raised and the problems noted were taken into account, all the necessary changes and comments were made. On my opinion, now the manuscript is suitable for publication without further revision.

·

Basic reporting

no comment

Experimental design

no comment

Validity of the findings

no comment